# Analysis of Conservation Priorities and Runs of Homozygosity Patterns for Chinese Indigenous Chicken Breeds

**DOI:** 10.3390/ani13040599

**Published:** 2023-02-08

**Authors:** Chaoqun Gao, Wenping Du, Kaiyuan Tian, Kejun Wang, Chunxiu Wang, Guirong Sun, Xiangtao Kang, Wenting Li

**Affiliations:** 1Henan Innovative Engineering Research Center of Poultry Germplasm Resource, College of Animal Science and Technology, Henan Agricultural University, Zhengzhou 450002, China; 2The Shennong Laboratory, Zhengzhou 450002, China

**Keywords:** Chinese native chicken, conservation priority, ROH island, genome-wide SNPs

## Abstract

**Simple Summary:**

The abundant animal genetic resources ensure sustainable and successful development of animal husbandry. The lack of appropriate long-term conservation programs, however, leads to the absence of rich genetic diversity in indigenous breeds. We investigated the population structure, genetic distance, contribution priority, and runs of homozygosity (ROH) patterns for eight Chinese indigenous chicken breeds using genomic data. Our analysis showed that Chahua, Xiaoshan, and Wannan three-yellow chickens, should be the top three breeds for conservation priority. Selection signals based on ROH were considered to be associated with meat production traits such as body weight, carcass weight, breast muscle weight, drumstick and thigh percentage.

**Abstract:**

To achieve sustainable development of the poultry industry, the effective conservation of genetic resources has become increasingly important. In the present study, we systematically elucidated the population structure, conservation priority, and runs of homozygosity (ROH) patterns of Chinese native chicken breeds. We used a high-density genotyping dataset of 157 native chickens from eight breeds. The population structure showed different degrees of population stratification among the breeds. Chahua chicken was the most differentiated breed from the other breeds (Nei = 0.0813), and the Wannan three-yellow chicken (WanTy) showed the lowest degree of differentiation (Nei = 0.0438). On the basis of contribution priority, Xiaoshan chicken had the highest contribution to the total gene diversity (1.41%) and the maximum gene diversity of the synthetic population (31.1%). WanTy chicken showed the highest contribution to the total allelic diversity (1.31%) and the maximum allelic diversity of the syntenic population (17.0%). A total of 5242 ROH fragments and 5 ROH island regions were detected. The longest ROH fragment was 41.51 Mb. A comparison of the overlapping genomic regions between the ROH islands and QTLs in the quantitative trait loci (QTL) database showed that the annotated candidate genes were involved in crucial economic traits such as immunity, carcass weight, drumstick and leg muscle development, egg quality and egg production, abdominal fat precipitation, body weight, and feed intake. In conclusion, our findings revealed that Chahua, Xiaoshan, and WanTy should be the priority conservation breeds, which will help optimize the conservation and breeding programs for Chinese indigenous chicken breeds.

## 1. Introduction

China is one of the earliest chicken domesticated regions, and it promotes the development of chicken breeds [1]. China has 115 native chicken breeds with abundant phenotypes, such as excellent disease resistance [2], high-quality meat and eggs [3], and extensive adaptability to harsh environmental conditions. These characteristics are very important for future poultry breeding. However, several Chinese native chicken breeds are currently under threat of extinction because of the introduction of exotic chicken breeds [2]. Hence, it is crucial to establish an optimal conservation strategy to protect these chicken genetic resources. The assessment of conservation priority is a useful technique for defining the conservation value of breeds in order to effectively use the limited conservation funds and manage the extant genetic diversity.

Several approaches have been developed to estimate the conservation values of breeds by using various methods (e.g., microsatellites and single nucleotide polymorphisms [SNPs]). The Weitzman approach [4], a representative method, has been widely used to analyze conservation priorities. This method considers only the effect of inter-breed diversity alone and ignores intra-breed diversity. Additionally, the effectiveness of the Weitzman approach is reduced by abnormally high frequency of rare alleles due to founder effects, inbreeding, or strict genetic isolation. Compared to the Weitzman approach, another approach reported by Cortés et al. and Nei mainly relies on intra-breed diversity [5,6]. In this approach, a breed is prioritized when its eradication from a population will result in the greatest reduction in global average heterozygosity or allele diversity. This approach does not consider inter-breed diversity. An alternative approach that combines intra-breed and inter-breed diversities was used to investigate the contribution of breeds to global diversity and to improve the accuracy of conservation priority analysis [7]. A marker-based inter- and intra-population kinship estimate is available for conservation priority assessment (i.e., core set approach), and this estimate differs from the Weitzman approach as it attempts to conserve founding populations (thereby minimizing the loss of alleles) [8]. To date, few systematic studies have been conducted on conservation priorities. For example, Glowatzki-Mullis et al. analyzed the conservation priority of Swiss goat breeds by using microsatellite markers with the Weitzman approach and the Caballero approach [9]. Ginja et al. and Liu et al. also used microsatellite markers with the alternative approach to estimate the genetic diversity and their conservation priorities for Iberoamerican cattle [10] and Chinese indigenous goats [11], respectively, with the alternative approach. In another study, the extinction probability, contribution rate, and marginal diversity of 21 Chinese domestic duck breeds were evaluated to help determine conservation priorities by using microsatellite markers to determine conservation priorities [12]. Conservation priority has also been calculated using SNPs in studies performed on the global taurine populations [5,13]. An earlier investigation focused on the genetic characteristics and conservation priorities of Plymouth Rock chicken breed from several chicken lines [14]. However, the conservation priority of Chinese native chicken breeds has rarely been rarely evaluated.

Conservation policies should prioritize the maintenance of genetic diversity so that the population can adapt to potential selection pressures in the future [15]. Gene and allele diversities are commonly used to estimate genetic diversity [16]. Among them, gene diversity is sensitive to selection [17], while allele diversity is sensitive to the bottleneck effect [7,18,19]. A combination of various evaluation methods can be used to reduce the deviation in estimating both types of diversities. Therefore, we estimated the contribution of each population by calculating the changes in gene and allele diversities after removing the population groups sequentially from the cluster [7,20].

A thorough understanding of the germplasm for native breeds can facilitate orderly conservation management [21]. Runs of homozygosity (ROH) refers to continuous homozygous segments along the genome, and they have been widely used to detect ROH islands. An ROH island can be used to evaluate genetic diversity to investigate demographic history and to screen positive selection signatures of poultry or livestock [22,23]. A previous study showed that the length and number of ROH are related to population history [24]. Under continuous directional selection, the genome content of one population can be reshaped, and the frequency of ROH across the genome can increase [23,25]. Genomic regions with a low heterozygosity can be efficiently detected using genome-wide SNPs [26]. Cendron et al. used genome-wide SNP data to scan the ROH islands of Italian native chicken breeds and identified genes associated with growth, meat quality, feed conversion, and immunity located on the ROH islands [21,27].

In the present work, we conducted a comprehensive analysis of conservation priorities and ROH patterns for eight Chinese indigenous chicken breeds by using genome-wide SNPs and identified candidate genes for relevant economic traits. These findings can serve as a reference to improve the ongoing conservation efforts.

## 2. Materials and Methods

### 2.1. Genotyping and Samples

The current study used 157 chickens from eight breeds, including Baier Chicken (Baier), Chahua Chicken (Chahua), Fighting Chicken (Fighting), Gushi Chicken (Gushi), Langshan Chicken (Langshan), Wannan three-yellow Chicken (WanTy), Wugu Chicken (Wugu), and Xiaoshan Chicken (Xiaoshan). Baier chicken, Chahua chicken, Langshan chicken and Wugu chicken are included in the list of national livestock and poultry genetic resources protection in China. The details of the chicken breeds are provided in Table 1 and Appendix A (see Appendix A). All the data were downloaded from the Synergistic Plant and Animal (SYNBREED) project (www.synbreed.tum.de, accessed on 1 December 2021.), in which genotypes were obtained using a high-density Affymetrix^®^ Axiom™ chicken array [28]. A total of 579,621 SNPs were annotated on Gallus_gallus-5.0 [29].

### 2.2. Data Filtering

First, we filtered out 76 duplicated SNPs along with 490 SNPs with unclear chromosome annotation, and only SNPs from 28 autosomes were considered; thus, 26,861 SNPs were removed from both sex chromosomes. SNPs were filtered using PLINK 1.9 [30] with the following filtering conditions: (1) individual call rate of ≥95% and (2) SNP call rate of ≥99%; a total of 443,352 SNPs were reserved for the subsequent genetic diversity and ROH analysis, linkage disequilibrium (LD) pruning of SNPs was performed using PLINK with the parameters “--indep-pairwise 50 5 0.2,” thus leaving 157,968 SNPs for Multidimensional Scaling (MDS) analysis and phylogenetic tree construction.

### 2.3. Population Structure and Genetic Diversity

The MDS was first calculated using PLINK, and the phylogenetic tree was constructed using MEGA X [31] with 1000 bootstrap iterations. The iTOL web server [32] was used to display the phylogenetic tree. ADMIXTURE v1.3 [33] was used to analyze the population structure, and a data set with K = 11 was set up. Metapop2 [34] was used to calculate the minimum genetic distance of Nei [35], the intra-breed genetic diversity index, gene diversity (HT) and allele diversity (AT), the contribution of each breed to the maximum gene diversity (H) and allele diversity (K) of the synthetic pool, and the average number of private alleles per locus in the breed. Intra-breed diversity indices included the following: fii: average coancestry between individuals; si: average self-incompatibility of individuals; dii: average Nei distance between individuals; and Gi: proportion of diversity between individuals. To estimate the maximum contribution of each population to gene diversity, total gene diversity (HT) was divided into average gene diversity intra-breed (HS) and average gene diversity inter-breed (DG). HS was determined as 1 minus the average of intra-breed co-ancestries, and DG was calculated as the average Nei’s genetic distance inter-breed. Similarly, total allelic diversity (AT) was divided into intra-breed (AS) and inter-breed (DA). AS was measured directly from the average allelic richness minus 1 of the breed by El Mousadik and Petit [36], while DA was derived as the average number of unique alleles in a subpopulation compared to that in the other subpopulations averaged over all possible subpopulation pairs [35]. Furthermore, in a subsequent simulation experiment, when the gene and allele diversities of the synthetic pool (N = 1000) composed of subpopulations reached a peak, the subpopulations contributed to the components of this pool [18,34,37].

### 2.4. ROH Identification

PLINK was used to estimate the ROH of each individual. The detection parameters were as follows: (1) the minimum length of the ROH fragment was 1 Mb; (2) the maximum number of ROH fragments was two deletions and one heterozygous genotype; (3) at least 100 consecutive SNPs were present; (4) SNP density was 0.01 SNP/kb, and (5) the maximum interval between continuous homozygous SNPs was 1 Mb. The proportion of the number of times that each SNP was involved in ROH composition in the population was determined, and an SNP region involving >30% SNPs was regarded as an ROH island [23]. SNPeff V5.1 [38] was used to annotate the ROH island region to obtain the candidate genes, and the functions of the related genes were determined through Gene Ontology enrichment (http://kobas.cbi.pku.edu.cn/kobas3/, accessed on 1 December 2021.) and compared with those given in the quantitative trait loci (QTL) database (https://www.animalgenome.org/cgi-bin/QTLdb/GG/index, accessed on 1 December 2021.) to analyze the potential function of the candidate genes.

### 2.5. Linkage Disequilibrium Decay and Effective Population Size

PopLDdecay [39] was used to calculate and visualize the chain imbalance decay for all groups with the parameters -bin1 500 -bin2 1100 -break 2000. GONE [40] was used to estimate the change in historical effective population size and the current generation effective population size for all populations. Previous studies have reported that the total linkage map of the chicken genome ranges from 2600 to 3800 cM [41], and the total genome size is approximately 1100 Mb [29]. Approximately 3 cM genetic distance is equal to 1 Mb physical distance; hence, we used the parameter of 1 Mb = 3 cM for estimation.

## 3. Results

### 3.1. Population Structure, Population Divergence Analysis, and Relatedness among the Eight Native Chicken Breeds

MDS and phylogenetic tree analysis were conducted to determine the population structures of the eight Chinese indigenous chicken breeds (Figure 1). According to MDS analysis results, individuals from the eight breeds were grouped into their respective clusters (Figure 1A). Chahua chicken, as a less domesticated breed, was separated from the other breeds in the first dimension (C1). The results of phylogenetic tree analysis were consistent with those of MDS analysis, with Chahua chicken and Fighting chicken showing the longest branch and WanTy chicken showing the shortest branch (Figure 1B). The admixture analysis indicated K = 7 (CV error = 0.510, Figure 1C) as the most likely number of genetically distinct populations for 157 samples, when K = 7, all breed has a clear genetic background except the WanTy breed (Figure 1D). When K = 2, Chahua was first isolated; when K = 3, Fighting, Gushi and Langshan were isolated; when K = 5, Baier and Wugu were isolated; when K = 7, Xiaoshan was isolated. WanTy showed more mixed blood at K = 7, and when K = 8, it was isolated. It is worth noting that, similar to the phylogenetic tree analysis, a mixed blood individual appeared in the Wugu chicken (Figure 1D).

Pair-wise Nei’s minimum genetic distance (*D*_Nei_) was calculated for the eight native breeds (Figure 2A). The highest difference was found between Chahua chicken and Fighting chicken based on *D*_Nei_ of 0.095; moreover, both Chahua chicken and Fighting chicken showed high differences with other breeds. The lowest differentiation was observed between WanTy chicken and Xiaoshan chicken based on *D*_Nei_ of 0.035. Moreover, WanTy chicken showed low differences with all other breeds, with *D*_Nei_ of <0.058 (Figure 2A). This result was supported by the central position of WanTy chicken as being relatively central in both the MDS and phylogenetic tree analyses. Moreover, one animal classified as Wugu chicken was positioned within the cluster of Xianshan chicken. Intra-species diversity parameters, including fii, si, dii, Gi, and αi, were assessed for each breed (Figure 2B). Among these parameters, fii and si showed an identical pattern of change. Similarly, dii and Gi showed a positive association. WanTy had the relatively high diversity between individuals with dii (0.153) and Gi (0.488), while Langshan had the lowest diversity based on its lowest dii and Gi values.

### 3.2. Contribution of the Eight Chicken Breeds to Gene and Allele Diversities

The conservation goal can be directly defined by maximizing the global gene and allele diversities. We therefore calculated the change in the global gene and allele diversity by sequentially removing breeds from the meta-breed and evaluating their contributions. A positive or negative contribution value of the breed was correspondingly reflected by the loss or gain in diversity that resulted from its removal, as described by Metapop2. As shown in Figure 3A, the removal of Xiaoshan chicken resulted in the largest loss to the total gene diversity (HT, 1.41%), followed by Chahua chicken (1.16%). Positive contribution from Xiaoshan and Chahua chicken differed because Xiaoshan exhibited higher intra-breed diversity (HS), whereas, Chahua showed the highest inter-breed diversity (DG, 1.57%). Chahua chicken was confirmed to be the most distinct breed based on its largest Nei’s distance (0.071, Figure 2A). The largest gain of HT was derived from Gushi chicken (−0.37%), followed by Fighting chicken (−0.28%). Negative contributions of Gushi chicken and Fighting chicken to HS (−1.31% and −1.26%, respectively) indicated that both populations had low intra-breed diversity. Interestingly, although WanTy showed the highest negative contribution to DG (−1.74%), it will be listed as the first breed to be protected according to intra-breed gene diversity (2.05%). This result can be explained by its lowest average Nei’s distance (0.0438) as compared to that for other breeds (Figure 2A). A similar analysis was performed using another metric, namely total allelic diversity (AT, Figure 3B). The removal of WanTy led to the highest positive contribution of AT (1.63%), followed by the removal of Xiaoshan (1.23%). The highest negative contribution to AT was still from Gushi (−0.37%) and Fighting chicken (−0.28%) because of their allelic diversity (AS, −0.98% and −0.71% for Gushi and Fighting chicken, respectively). Chahua was ranked first on the basis of its contribution to HT, but it ranked fourth in AT because of its negative contribution to AS (−0.53%).

We then simulated the contributions of breeds to the synthetic pool of N = 1000 individuals and computed their proportion at the maximum value of the expected heterozygosity and the total number of alleles. As shown in Figure 3C, the first three breeds to contribute to genetic diversity were Xiaoshan chicken (31.3%), WanTy chicken (23.70%), and Chahua chicken (18.1%). The top three breeds to contribute to allelic diversity were WanTy chicken (17.0%), Xiaoshan chicken (16.0%), and Wugu chicken (12.6%). This finding was in accordance with the above-mentioned results for HT and AT. Additionally, a worthwhile observation is noting that Xiaoshan chicken retained the largest number of private alleles (Figure 3D), followed by WanTy chicken and Chahua chicken; this finding indicated that gene flow was more restricted in these breeds.

### 3.3. ROH Analysis Results

ROH were identified in the entire genome of the chicken breeds. ROH analysis revealed a total of 5242 ROH fragments, with the longest ROH fragment reaching 41.51 Mb. The distribution and detailed statistics of ROH are shown in Figure 4 and Table 2. The average ROH number (NROH) and the average length of individual ROH fragments (MNROH) across the eight chicken breeds were 653 and 2.654 Mb, respectively. Additionally, the maximum and minimum NROH were 975 and 91 for the individuals corresponding to WanTy chicken and Fighting chicken, respectively. The number of ROH on each chromosome was positively correlated with chromosome size (Figure 4B).

As shown in Figure 5, from the total sample, we identified five ROH islands on GGA1 (73.2–75.88 Mb), GGA2 (51.1–54.05 Mb), GGA5 (1.91–3.91 Mb), GGA8 (8.95–11.68 Mb), and GGA11 (2.37–3.77 Mb); these five ROH islands were shared by >30% of chickens, and the islands together covered 11.76 Mb of the entire genome. These five ROH islands contained 135 annotated genes, including 80 protein-coding genes, which were involved in growth, feed conversion rate, abdominal fat ratio, and other traits (Appendix A).

Interestingly, 69.43% of the individuals in the population shared the ROH island on GGA5 (Figure 5). Annotated genes from this ROH were enriched in amino acid metabolism, reproduction and neural development, ion transport across the membrane, and microtubule production (Appendix A).

In addition, we compared the overlapped regions of the ROH islands to the animal quantitative trait loci (QTL) database. Table 3 provides detailed information of their overlapped region. The overlapped region on GGA5 was related to production traits such as body weight, carcass weight, head width, and protein height. The QTL regions on the other four chromosomes were associated with immunity, carcass weight, drumstick and leg muscle development, egg quality and egg production, abdominal fat precipitation, body weight, and feed intake (Appendix A).

The QTL-related traits were further classified into meat, egg, fat, immunity, and other traits (Figure 6) (see Appendix A for detailed classification). QTLs related to meat traits accounted for 44.00% of the total QTLs, indicating that these regions played an important role in chicken muscle growth and development. This finding provides a theoretical reference for the breeding direction of indigenous chicken breeds.

### 3.4. LD Decay and Effective Population Size (Ne)

The patterns of LD decay patterns were expected to be consistent with ROH results. For instance, for WanTy chicken (orange line), we observed a high level of expected heterozygosity, the smallest LD distance, and a rapid LD decay (Figure 7). Similarly, for Gushi chicken (purple line), a low level of expected heterozygosity was observed according to the ROH result, and Gushi chicken showed the largest LD values and an overall slow LD decay.

The effective population size (Ne) was estimated across the entire genome (Figure 8). As shown in the historical trace of Ne for the past 727 generations, all the populations showed a sharp decline in the Ne value in the past 450 generations, except that the WanTy chicken initially showed a mild decline and then exhibited a progressive in the Ne value increase in the past 500 generations. The Ne value for the current generation was in agreement with ROH and LD decay patterns. The WanTy chicken presented largest Ne with more than 300, and Fighting chicken was the smallest one with less than 30.

## 4. Discussion

The primary goal of a conservation strategy is to maintain a high level of genetic and allelic diversities in a population [18]. Generally, to protect the germplasm characteristics of livestock and poultry, it is necessary to evaluate the population diversity and to reconstruct livestock history [42]. China has abundant chicken genetic resources, which are assumed to be important and unique gene resources because of the absence of severe selection pressure for these breeds. Hence, in our present study, we systematically analyzed the population structure and the contribution of different breeds to genetic diversity and ROH islands to provide a reference for the conservation priority of indigenous chicken breeds in China.

The population structure analysis revealed that Chahua chicken was the most differentiated breed (Figure 1 and Figure 2). Thus, this breed might have positively contributed to the population diversity of Chinese indigenous chickens in terms of gene diversity or allele diversity. As reported previously, Chahua chicken has an older matrilineal haplotype, and it is a transitional breed between domestic chicken and red jungle fowl [43]. This is also proved by the fact that Chahua chicken was first isolated in the Admixture analysis. This could be a reason why Chahua contributed relatively more to the overall population. It should be noted that in the phylogenetic tree and Admixture analysis, a Wugu chicken individual appeared in the Xiaoshan chicken cluster, which may imply gene introgression or sample contamination. Furthermore, according to the results of the two estimation methods, we found that Xiaoshan and WanTy have always contributed greatly diversity of the entire population because of their lower kinship and greater Nei’s genetic distance (Figure 2B) [18]. In contrast, Fighting chicken and Gushi chicken had the lowest contribution to the population diversity because of their higher between-individual kinship [44]. Interestingly, the lower kinship of WanTy chicken might be due to gene flow or hybridization, as shown in the LD decay pattern and historical trace of the Ne (Figure 7 and Figure 8) [45]. Gushi chickens are present in a relatively isolated area, with less opportunity for genetic exchange with other chicken populations, while Fighting chickens, because of their special use, are subjected to the inbreeding process [45,46].

Another well-known technique for identifying the loss of genetic diversity within breeds is ROH [23]. Regions of overlapping homozygosity that are highly shared among individuals in a population are called ROH islands. As directed artificial selection reduces genomic variability, ROH islands have been considered as potential markers of selection around target loci [23,47,48]. Therefore, the identification of ROH islands is helpful to find the variation of target traits, which will be beneficial to the development and utilization of animal resources, and then promote the conservation of animals [21]. Five ROH islands were detected in this study. The existence of ROH islands indicates that the genome has been affected by selection, and there might be candidate genes related to breed-specific characteristics (Appendix A). ROH signals were significantly enriched in stress, immunity, and lipid metabolism pathway, for example, *ANO5*, *NELL1*, and *BBOX1* genes were located on chromosome 5 and *NFATC3* and *ESRP2* genes were located on chromosome 11. The *ANO5* gene is involved in the development of muscle tissue and estrogen production in mice [49]. The *NELL1* gene is an important growth factor related to bone tissue formation and bone integrity, and it is usually expressed in commercial broilers to achieve a high growth rate and meat yield [50]. The *BBOX1* gene, which is involved in the regulation of feed efficiency, is highly expressed in high-growing commercial chickens [51]. The *NFATC3* gene [52] is an important member of the NFAT family and plays a critical role in the transformation of muscle fiber types. The *ESRP2* gene [53] is associated with the abdominal fat content in chickens. A comparative analysis of the overlapping genomic regions between the ROH islands and the the QTLs in the QTL database (Appendix A) revealed that the ROH islands overlapped with economic traits-related QTLs, including body weight, carcass weight, immunity, egg quality and egg production, feed intake, and feed conversion ratio.

Collectively, our study indicated that the eight native chicken breeds had moderate Ne, except for WanTy. This implies that the conservation program should be optimized further. The effectiveness of the conservation program should be estimated annually by using genomic data for real-time monitoring of the status of chicken genetic resources.

## 5. Conclusions

The present study comprehensively evaluated the conservation priorities of eight indigenous chicken breeds in China by using genomic data. Chahua, Xiaoshan, and WanTy chicken breeds should be given a high priority in conservation programs. The ROH islands revealed that the selection targets were associated with meat-production traits, thus implying that Chinese native chickens have the potential for meat-type breeding. Taken together, our findings can be used to improve conservation strategies for chickens and have practical relevance for chicken genetic resource conservation in China.

## Figures and Tables

**Figure 1 animals-13-00599-f001:**
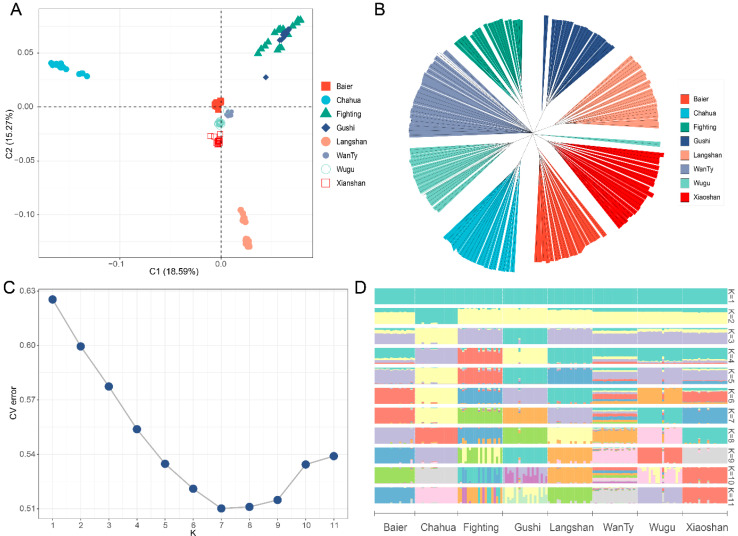
(**A**) MDS analysis and (**B**) phylogenetic tree of the eight indigenous breeds, (**C**) cross-validation (CV) error rate and (**D**) population structure of the eight indigenous breeds.

**Figure 2 animals-13-00599-f002:**
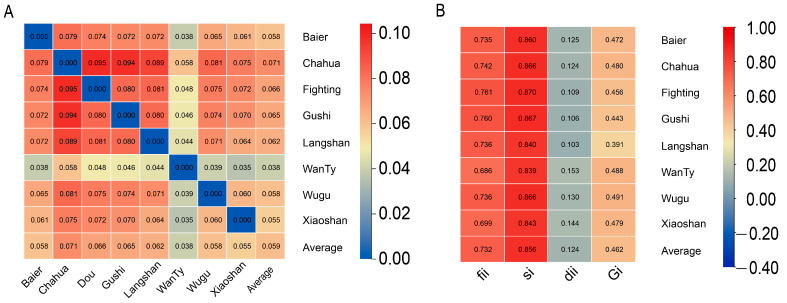
(**A**) Nei’s minimum genetic distance inter-breed and (**B**) intra-breed diversity indices.

**Figure 3 animals-13-00599-f003:**
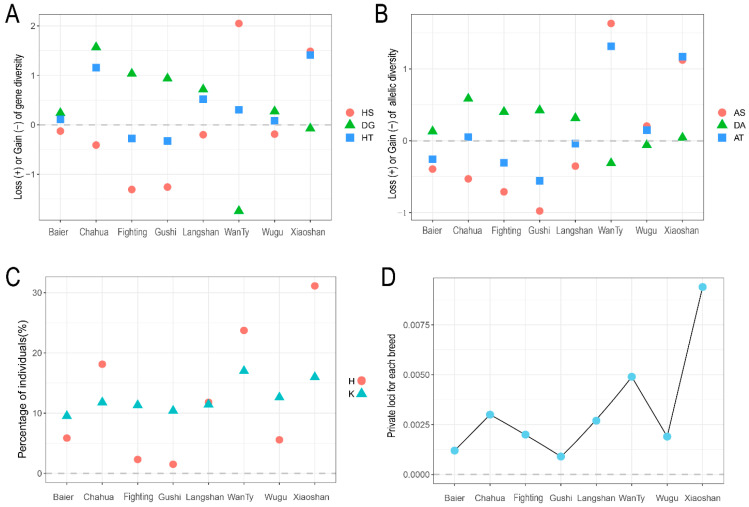
Contribution of the eight native chicken breeds to genetic diversity. (**A**) Loss (+) or gain (−) of genetic diversity after sequential removal of each breed. HS is the gene diversity intra-breed, DG is the gene diversity inter-breed, and HT is the total loss or gain of gene diversity. (**B**) Loss (+) or gain (−) of allelic diversity after sequentially removal of each breed. AS is allelic diversity intra-breed, DA is allelic diversity inter-breed, and AT is the total loss or gain of allelic diversity; (**C**) contribution of individuals from each breed to pools with the greatest proportion of genetic diversity (H) and allelic diversity (K) proportion. (**D**) Average number of private alleles per locus in each breed.

**Figure 4 animals-13-00599-f004:**
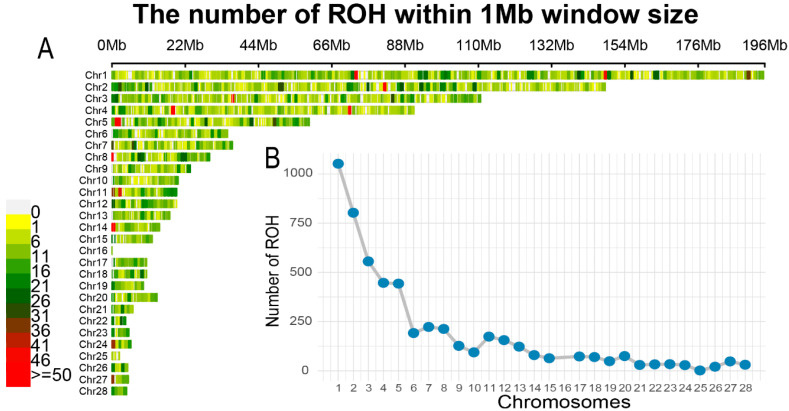
(**A**) Number of ROH within 1 Mb window and (**B**) number of ROH on each chromosome.

**Figure 5 animals-13-00599-f005:**
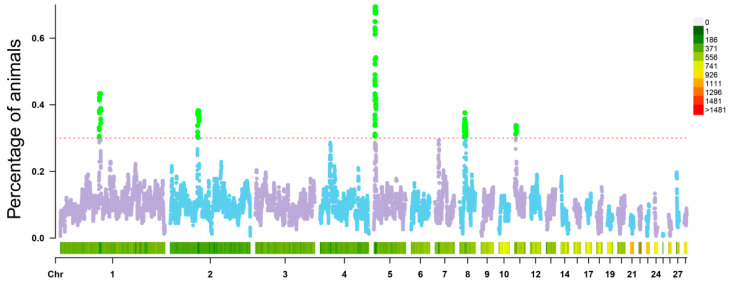
ROH islands across the entire genome.

**Figure 6 animals-13-00599-f006:**
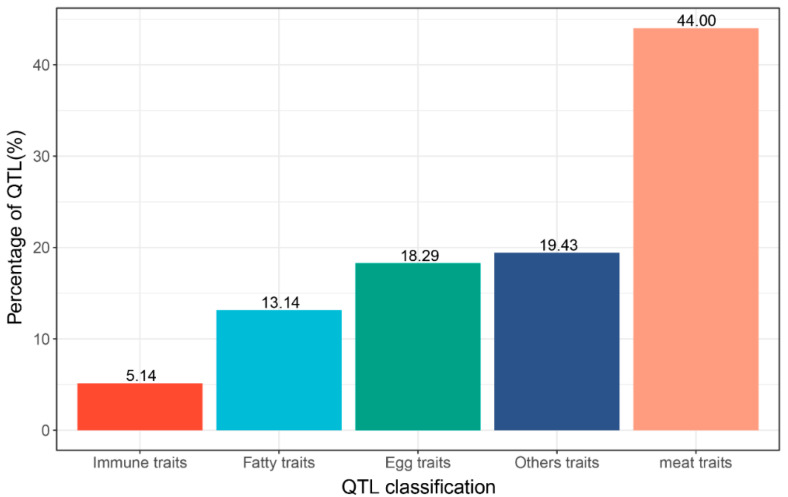
Statistics of QTL classification.

**Figure 7 animals-13-00599-f007:**
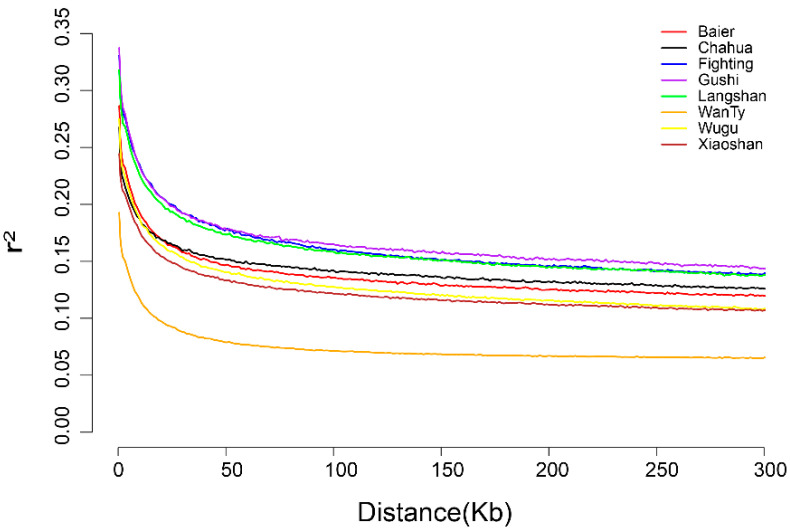
Linkage disequilibrium (LD, r2) decay for the eight Chinese native breeds. Genetic distance is measured in kilo base pair (kb).

**Figure 8 animals-13-00599-f008:**
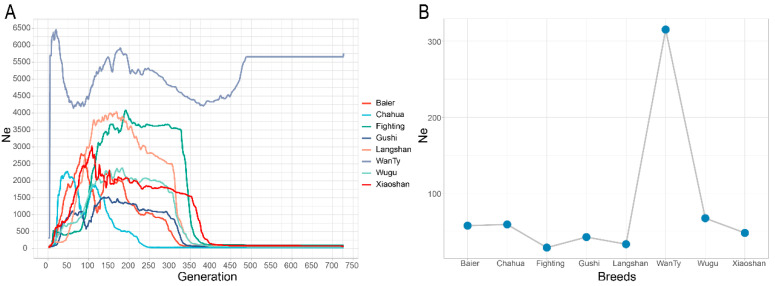
Historical trace of the effective population size (Ne) from the genomic data for the past 727 generations (**A**) and the effective population size at the current generation of the eight indigenous breeds (**B**).

**Table 1 animals-13-00599-t001:** Breed information.

Breeds	Label	Number	Original Region	Specific Features
Baier Chicken	Baier	18	Shangrao city, Jiangxi	Light-sized, three yellow, layer
Chahua Chicken	Chahua	19	Xishuangbanna, Yunnan	light-sized, meat and egg dual-purpose breed
Figthing Chicken	Figthing	20	Zhengzhou city, Henan	Heavy-sized of purpose breed, fancy breed
Gushi Chicken	Gushi	20	Gushi county, Henan	Medium-sized, Three yellow, meat and egg dual-purpose breed
Langshan Chicken	Langshan	20	Zhengzhou city, Henan	Heavy-sized, meat and egg dual-purpose breed
Wannan three-yellow chicken	WanTy	20	Qinyan county, Anhui	Medium-sized, three yellow, egg purpose breed
Wugu Chicken	Wugu	20	Taihe county, Jiangxi	Light-sized, White feather, black skin, black bone, medicinal and meat
Xiaoshan Chicken	Xiaoshan	20	Taihe county, Jiangxi	Heavy-sized, meat and egg dual-purpose breed

**Table 2 animals-13-00599-t002:** Statistics of ROH.

Breed	Sample Size	SROH (Mb)	NROH	MNROH ± SD (Mb)
Baier	18	1977.998	686	2.845 ± 0.645
Chahua	19	2698.555	837	3.227 ± 0.405
Fighting	20	2328.759	975	2.353 ± 0.401
Gushi	20	2745.876	886	3.006 ± 0.643
Langshan	20	571.271	296	1.921 ± 0.303
WanTy	20	181.114	91	2.010 ± 0.692
Wugu	20	2645.142	917	2.885 ± 0.576
Xiaoshan	20	1775.372	554	2.989 ± 0.724
Average	-	1865.511	653	2.648 ± 0.724

Note: SROH: Total length of all individual ROH fragments; NROH: Number of ROH fragments in all individuals; MNROH: Average length of individual ROH fragments.

**Table 3 animals-13-00599-t003:** Annotation results and ROH islands.

Chromosome	Number of SNPs	Start (bp)	End (bp)	Number of Genes	Number of QTL
1	469	73,276,840	75,884,166	18	55
2	671	51,100,728	54,046,793	13	30
5	400	1,912,343	3,909,043	18	16
8	638	8,946,537	11,679,072	11	54
11	343	2,365,463	3,771,468	20	55

## Data Availability

The raw data used in this study are publicly available, and can be obtained upon reasonable request to the corresponding author.

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
