# Peer review of "Analysis of Conservation Priorities and Runs of Homozygosity Patterns for Chinese Indigenous Chicken Breeds"

_animals, 2023, doi:10.3390/ani13040599_

Round 1

Reviewer 1 Report

Dear Authors, thank you for your work. I highlighted some points and added my comments on the attached pdf. Conservation and safeguarding of genetic resources are vital. For this reason, I would like to state that I find your work important. However, some points must be clarified:

- I was hoping to see conservation priorities when I started reading your article. But is there an existing conservation program on these breeds? Do the studied populations belong to conservation flocks? I did not see this information. Please give a detailed information about the population and recent conservation schemes of the breeds.

- It would be nice to give some photos of the breeds with a supplementary or   adding them to the phylogenetic tree figure.

- Although the information in the title and introduction of the article focused on ROH, the results could not be adequately interpreted in the discussion section. The discussion part is presented more like a final report on genetic similarity and allelic richness of breeds. Therefore, it would be appropriate to rewrite the discussion section on the ROH islands, their importance and their importance in conservation programs.

- Again, according to the results of the research, I would expect some populations to be prioritized especially in conservation programs, and some special conservation programs to be recommended for some breeds according to the ROH results. However, I did not find this information in the discussion part or the conclusion part, and in my opinion, these points will be the most valuable results of the study. Please reconsider this interpretation in your discussion section.

Author Response

Dear reviewer

Thank you for your suggestion, which makes our work more perfect.

All points requiring clarification have been answered.

Best regards

Reviewer 2 Report

Dear Sir

The article may be corrected as suggested in the file

Author Response

(The authors gave the same response as above.)

Reviewer 3 Report

This paper addresses conservation status of 8 native chicken breeds of China.

Several points need to be clarified before the paper can be accepted :

-          the origin of the breeds, and particularly their geographic location and selection history must be given. Table S1 provides unsufficient information, total population size should be given ; also the ROH analysis leads to suggest that selection took place on some genes important for meat production, does it match with the real story ? What is the growth rate, or adult body weight of these chicken breeds ?

-          furthermore the authors have decided to apply a method recommended for subdivided populations,t here there severals breeds and we do not understand whether gene flow can take place between them at present of not ? A synthetic population is mentioned in line 129 and a meta-breed is mentioned in line 191, without explanation. The authors must explain what they mean by this. Generally a synthetic populations results from the cross of several breeds, but crossbreeding is not mentioned at all in the paper.

-          the authors should explain how they define an allele for the calculation of AS, reference 17 explains that one allele is a haplotype of 5 consecutives SNPs, so that the number of alleles can reach 32 per locus, is the same definition used here ?  (in section 2.3)

-          the simulation experiment mentioned lines 84-87 has been done in reference 22 but the link with the present results is unclear, why do the authors wish to say here ?

-        Finally, the conservation priorities are not well justified, the conclusion mentions 3 breeds, including the WanTy, which currently has a large Ne, so that the conservation program for WanTy is likely to differ from the program to be applied for Chahua and Xiaoshan, which have a much smaller Ne. This point should be addressed in the discussion, it would increase the scientific value of the conclusion.

There are a number of mistakes in the vocabulary, 

-          line 74 ‘Plymouth Rock children’ should be Plymouth Rock chicken

-          line 88  ROH means « Runs of homozygosity » and not « Runs of homogeneity »

-          ‘homogeneity’ is used elsewhere in the text (for example line 131), the authors should check the meaning, what do they want to say ?

-          line 90 ROH are studied ‘along the genome’ rather than ‘across the genome’

-          line 118 : individual ‘deletion rate’ <= 5% does not mean anything, I suppose that the authors wanted to say ‘detection rate’, please correct

-          line 111 : is it said that the data were downloaded from Synbreed project, the authors must provide the link to the data (url address), all the more that they say these data are publicly accessible

-          lines 143-144 : ‘the maximum number of ROH fragments was 2 deletions and 1 heterozygous genotype’ ?? what does this mean ? why 2 deletions ? where are the deletions ?

-          lines 214-223 : was simulation needed to reveal contributions ? the results already show that Wanty was more variable , why 1000 individuals when you only have 157 with real data ?

-          line 216 : figure 3C and not 3D

-          lines 313- 316 : contradiction, line 313 mentions a lower kinship for Xiaoshan and WanTy and line 316 mentions a higher kinship of WanTy, please check which breed has to be mentioned on line 316, WanTy is likely a mistake

Figure 1 : please provide the % of variance for C1 and for C2 ; breed names are not exactly the same on Fig 1A and 1B : Xiangshan in 1A and Xiaoshan in 1B, please correct

Figure 3 : legend, line 231 : why do we see the word ‘cultivar’ ? and what are the ‘pools’ ; this lets suggest that this figure legend comes from another paper ?

Table 1 : WanTy breed has a very small total length of ROH, which is consistent with the fact that it is the most diverse, but Langshan has the second smallest total length of ROH, while this breed was found to exhibit the lowest diversity (line 183), generally a breed with a low diversity has a larger total length of ROH than breeds which are more diverse, such as Xiaoshan

Author Response

(The authors gave the same response as above.)

Reviewer 4 Report

I also think that it is important that several domestic breeds of chickens are examined in terms of genetic diversity to protect genetic resources.

The method in this manuscript is fine.

However, when the authors want to see the genetic diversity not only intra-breed but also inter-breed at the same time, I recommend to apply "Admixture" analysis.

It reveals not only the sharing genetic backgrounds among eight breeds but also backgrounds shared by a few populations and each population specific component.

For effective population size estimation, the authors use "GONE" based on linkage disequilibrium. However, Figure 8B showed the present effective size is too small. For the Fighting chicken, Ne=30, then the nucleotide diversity is 4X Ne X u x g = 120X 1.9 X 10^-9 X 1 = 2.3 x 10^-7. Since the genome size of chicken is 1.2 X 10^9 bp, the expected number of differences is 276. Is within Fighting chicken variability so small?  It is OK that recent demographic history is estimated from the linkage disequilibrium. However, the estimation of extant effective population size by GONE is different from the estimate of authentic effective population size.  The extant effective population size should be estimated from the nucleotide diversity, 4Neug, where u is the mutation rate per site per generation and g is the generation time. 

Author Response

(The authors gave the same response as above.)

Round 2

Reviewer 4 Report

Thank you very much for the author's reply.

The authors address adequately my comments.

I found that several Ne estimation methods differ from each other by focusing time.

For example, Ne based on LD is for quite recent Ne, whereas Ne based on nucleotide diversity is based on more ancient information and is Ne for longer time. In this manuscript, small Ne based on LD may reflect a recent inbreeding in the chicken breed.